# A Technical Report on "Erasing the Invisible": The 2024 NeurIPS Competition on Stress Testing Image Watermarks

**Mucong Ding**[1*], **Tahseen Rabbani**[1*], **Bang An**[1*], **Chenghao Deng**[1*], **Anirudh Satheesh**[1],
**Souradip Chakraborty**[1], **Mehrdad Saberi**[1], **Yuxin Wen**[1], **Kyle Sang**[1], **Aakriti Agrawal**[1],
**Xuandong Zhao, Mo Zhou, Mary-Anne Hartley, Lei Li, Yu-Xiang Wang,**
**Vishal M. Patel, Soheil Feizi**[1],**Tom Goldstein**[1*], **Furong Huang**[1*]

[1]University of Maryland
*{mcding,trabbani,bangan,dengch16,tomg,furongh}@umd.edu

## Abstract

AI-generated images have become pervasive, raising critical concerns around content authenticity, intellectual property, and the spread of misinformation. Invisible watermarks offer a promising solution for identifying AI-generated images, preserving content provenance without degrading visual quality. However, their real-world robustness remains uncertain due to the lack of standardized evaluation protocols and large-scale stress testing. To bridge this gap, we organized "Erasing the Invisible," a NeurIPS 2024 competition and newly established benchmark designed to systematically stress testing the resilience of watermarking techniques. The competition introduced two attack tracks—Black-box and Beige-box—that simulate practical scenarios with varying levels of attacker knowledge on watermarks, providing a comprehensive assessment of watermark robustness. The competition attracted significant global participation, with 2,722 submissions from 298 teams. Through a rigorous evaluation pipeline featuring real-time feedback and human-verified final rankings, participants developed and demonstrated new attack strategies that revealed critical vulnerabilities in state-of-the-art watermarking methods. On average, the top-5 teams in both tracks could remove watermarks from $\geq 89\%$ of the images while preserving high visual quality, setting strong baselines for future research on watermark attacks and defenses. To support continued progress in this field, we summarize the insights and lessons learned from this competition in this paper, and release the benchmark dataset, evaluation toolkit, and competition results. "Erasing the Invisible" establishes a valuable open resource for advancing more robust watermarking techniques and strengthening content provenance in the era of generative AI.

## 1 Introduction

Recent advances in text-to-image generation have captivated the AI community and the general public alike. Open-source models like Stable Diffusion and proprietary models such as DALL-E, Midjourney, and GPT-4o allow users to create images that are virtually indistinguishable from those crafted by humans. This surge in AI-generated content has prompted the AI/ML community, as well as policymakers, to focus on developing mechanisms to identify and attribute AI-generated content - as highlighted in recent guidance from the EU AI Act and Executive Office of the President.

Watermarks offer a promising solution to identify AI-generated images. A watermark is a signal embedded in an image to signify its origin or ownership, ideally without degrading image quality.

39th Conference on Neural Information Processing Systems (NeurIPS 2025) Track on Datasets and Benchmarks.

Modern watermarks, especially those invisibly embedded within images by generative AI models, are designed with the best intentions. However, they are not without their pitfalls.

**Pitfall I**  A primary concern is the false sense of security they may impart. Users might believe that a watermark's detection accuracy, largely measured by the recall rate of watermarked instances, is reliably high. But what happens when these images are slightly altered using readily available editing tools? Can watermarks withstand such modifications?

Despite the robustness of many watermarks against common image manipulations, determined attackers can sometimes succeed in removing them. Unfortunately, a lack of standardized evaluations in the literature and the absence of large-scale stress tests have led to an incomplete understanding of the true robustness of these watermarking techniques.

**Pitfall II**  Furthermore, ensuring the precision remains high is vital to avoid false positives, where genuine, non-AI-generated images are mistakenly flagged as synthetic. Some researchers are concerned that the damage from a false positive, incorrectly accusing someone of using generative AI, could be much greater than that of a false negative, where an AI-generated image goes undetected. Therefore, the robustness to spoofing attacks [Saberi et al., 2024] is also important.

In this work, we address Pitfall I by conducting a large-scale, standardized stress test to rigorously evaluate watermark robustness against removal attacks. In this vein, WAVES (Watermark Analysis via Enhanced Stress-testing) [An et al., 2024] introduced a benchmark with a standardized evaluation protocol and 26 attacks, ranging from classical image distortions to sophisticated adversarial methods, to systematically probe watermark robustness. Although WAVES revealed notable vulnerabilities in several watermarks, its attack suite remained relatively small-scale, limiting the breadth of insights into real-world threats. To overcome this limitation and to gather the most potent removal techniques available, we extended WAVES by organizing Erasing the Invisible (ETI), a global NeurIPS competition that serves as a large-scale stress test for state-of-the-art watermarking methods.

The ETI competition comprises two tracks, Black-box and Beige-box, that emulate practical deployment scenarios with varying levels of attacker knowledge. The Black-box track mirrors industry settings where watermark algorithms are proprietary and undisclosed, challenging participants to disrupt unseen watermarks while preserving image fidelity. The Beige-box track, by contrast, provides labels on the watermarking methodology, enabling attackers to tailor their strategies to known embedding processes. Attracting 2,722 submissions from 298 teams worldwide, ETI employed a rigorous, standardized evaluation pipeline complete with a real-time leaderboard and human-verified final rankings. Through this large-scale effort, we uncovered new vulnerabilities. Top teams succeeded in removing $\geq 89\%$ of watermarks on average without perceptible quality loss. The innovative attack strategies serve as strong baselines for future research. To drive continued progress in digital content provenance, we summarize the insights and lessons learned in this paper and open-source the ETI benchmark, including the dataset, evaluation toolkit, and complete competition results. Moreover, the competition pipeline can be effortlessly extended to live or rolling benchmarks, enabling continual evaluation of emerging watermarking techniques.

## 2 Recap of the Competition

### 2.1 Competition Structure and Design

Images for the competition were generated via a hybrid approach using both the **Flux. 1 [dev]** model and **Stable Diffusion 2.1**. We incorporated watermarking methods spanning two fundamental paradigms: *in-processing* techniques (embedding the watermark during the image generation process) and *post-processing* techniques (applying the watermark to already generated images). To establish a standardized baseline, we calibrated the false-positive rate (FPR) for each deployed watermark. For every decoder, we computed continuous detection scores on 10,000 unwatermarked images and set the decision threshold to yield an **FPR of 0.1%**; this threshold was then held fixed for all subsequent evaluations. Under this fixed threshold, all watermarking methods achieved a true-positive rate (TPR) of at least 99.9% on clean, un-attacked watermarked images. This provided a consistent foundation for assessing participants' removal efficacy. The competition featured two distinct tracks, differing in the level of knowledge afforded to attackers regarding the employed watermarks.

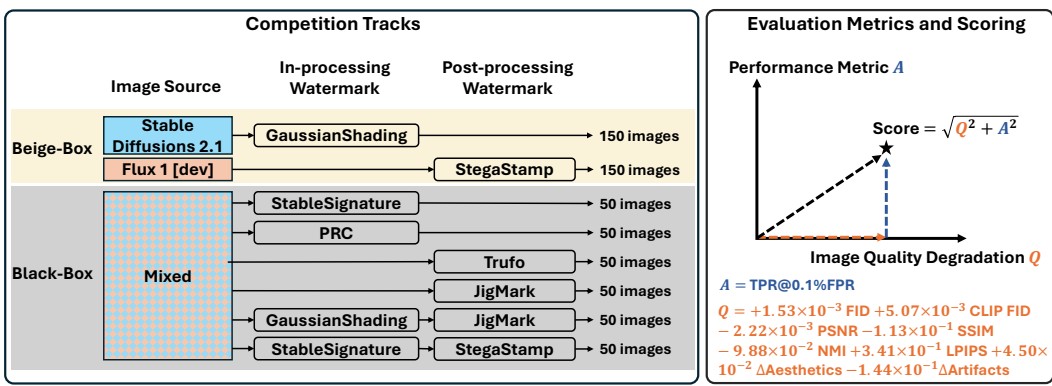

Figure 1: Overview of competition design, including track structure (Left) and evaluation metrics (Right).

**Beige-box Track.** In this track, participants were informed of the specific watermarking algorithms applied to the images, facilitating more targeted attack strategies. Two representative methods were employed, distributed across 300 images: (1) **Gaussian Shading** [Yang et al., 2024], an *in-processing* watermark, was applied to 150 images generated by Stable Diffusion 2.1. This method embeds a unique pattern within the initial noise vector integral to the diffusion model's synthesis process. (2) **StegaStamp** [Tancik et al., 2020], a *post-processing* watermark, was applied to 150 images generated by Flux. 1 [dev]. It utilizes a deep neural network trained for robust, invisible watermark embedding via learned encoding and decoding.

**Black-box Track.** Participants in this track received 300 images with a more complex and entirely confidential mix of watermarks: (1) **Single Watermark**: Four sets of 50 images were each watermarked using one distinct method: JigMark [Pan et al., 2024], PRC [Gunn et al., 2023], StableSignature [Fernandez et al., 2023], or the Trufo watermark. (2) **Double Watermark**: 50 images featured a combination of Gaussian Shading + JigMark, and another 50 combined StableSignature + StegaStamp. All images within this track were shuffled to prevent participants from inferring watermarking methods based on image order or batching.

### 2.1.1 Evaluation Metrics and Scoring.

To rigorously evaluate the trade-off between watermark removal success and image quality preservation, we implemented a sophisticated evaluation system based on principles from the WAVES benchmark [An et al., 2024]. Each participant submission was assigned a final score, calculated as the Euclidean distance in a 2D space defined by two core metrics: **image quality degradation** ($Q$) and post-attack **watermark detection performance** ($A$). We set Score = $\sqrt{Q^2 + A^2}$. The primary objective for participants (attackers) was to **minimize this score**. A low score signifies an effective attack that significantly degrades or removes the watermark (low $A$) while incurring minimal perceptible damage to image quality (low $Q$).

**Image Quality Degradation** ($Q$). The visual impact of an attack was quantified by comparing each attacked image to its original watermarked (but unattacked) version. The normalized score $Q$ aggregated information from eight distinct image quality metrics (IQMs): (1) Peak Signal-to-Noise Ratio (PSNR), (2) Structural Similarity Index (SSIM) [Wang et al., 2004], (3) Normalized Mutual Information (NMI), (4) Frechet Inception Distance (FID) [Heusel et al., 2017], (5) CLIP Image Fidelity (CLIP-FID) [Kynkäänniemi et al., 2019], (6) Learned Perceptual Image Patch Similarity (LPIPS) [Zhang et al., 2018], (7) Delta Aesthetics Score ($\Delta$Aesthetics) [Xu et al., 2023], and (8) Delta Artifacts Score ($\Delta$Artifacts) [Xu et al., 2023]. A lower $Q$ indicates less visual degradation. Details on IQM normalization and weighting are provided in the Appendix.

**Watermark Detection Performance** ($A$). A critical aspect of our evaluation was the precise and efficient calculation of the post-attack **watermark detection performance** ($A$). To achieve this, for each distinct watermarking method (including superpositions), we first pre-calculated a specific *detection score threshold* for its decoder. This threshold was carefully determined by evaluating the decoder on a large, held-out set of 10,000 diverse unwatermarked images (per watermark type/superposition) and

identifying the raw detection score that corresponded to a False Positive Rate (FPR) of 0.1%. This pre-calibration provided an accurate estimation of the 0.1% FPR operating point and enabled efficient computation of $A$ for incoming participant submissions (each typically containing 300 images). The metric $A$ for a submission was then defined as the True Positive Rate (TPR) achieved by the original watermark detector on the participant's attacked images, using this pre-calibrated 0.1% FPR score threshold; thus, $A = \text{TPR@0.1\%FPR}$. Consequently, $A$ represents the fraction of attacked images where the watermark is still successfully detected. For an attacker, a *lower value of $A$ is desirable*, as it signifies that their attack has rendered the watermark undetectable in a larger proportion of images. An $A$ value approaching 0 indicates near-complete success in watermark removal, while an $A$ value approaching 1 suggests the watermark largely withstood the attack.

### 2.1.2 Competition Platform and Deliverables.

The competition was hosted on *Codabench* [Farragi et al., 2020–], an open-source platform for computational challenges. The core evaluation logic was implemented in an open-source Python program and docker image open sourced at *Github* [1][2], executed by custom, containerized compute workers built upon the standard Codabench architecture. For each submission of attacked images, the evaluation pipeline automatically performed watermark decoding and image quality assessment (following verification and standardized preprocessing). It then computed the final score from $Q$ and $A$, reporting it to the real-time Codabench leaderboard. Key deliverables for future research include the original watermarked image datasets and the complete set of participant submissions with detailed evaluation results, all publicly released on *HuggingFace* [3].

## 3 Major Insights via Result Analysis

### 3.1 Submission Statistics Overview and Understanding

The competition ran from September 16 to November 5, 2024, attracting significant global engagement. A total of **2,722 submissions** were received across both tracks. Specifically, the **Beige-box Track** saw **1,072 submissions** from **65 distinct teams**, while the **Black-box Track** recorded **1,650 submissions** from **77 distinct teams**. This broad participation highlights the community's strong interest in the challenge of evaluating and breaking image watermarks. Figure 2 provides a visual summary of key submission statistics and outcomes.

**Varied Robustness of Black-Box Watermarks.** Figure 2a presents the distribution of watermark detection rates for the six distinct watermark algorithms (GaussianShading, JigMark, PRC, StableSignature, StegaStamp, and Trufo) within the Black-box track, aggregated across all participant submissions. These results offer a broad overview of each method's resilience when subjected to a diverse array of attack strategies, ranging from simple edits to more sophisticated techniques. Notably, GaussianShading exhibited the highest median detection rate, suggesting it was, on average, the most challenging watermark for participants to successfully remove or degrade across all submissions (i.e., attacks resulted in higher residual watermark detection performance). Conversely, StableSignature and Trufo displayed the lowest median detection rates, indicating they were more frequently compromised (i.e., attacks achieved lower watermark detection performance). This aggregation reflects how these watermarks fare against a wide spectrum of attacker efforts, including potentially less polished or casual attempts.

**Watermark Detection Performance versus Image Quality Degradation Trade-offs.** The scatter plots in fig. 2b illustrate the crucial trade-off between watermark detection performance (axis $A$, lower is better for an attacker, signifying more effective watermark removal) and image quality degradation (axis $Q$, lower is better for an attacker, signifying better preservation of image quality) for submissions in both the Beige-box (left panel) and Black-box (right panel) tracks. The red lines delineate the Pareto frontiers, representing the best-achieved compromises by participants.

For the Beige-box track, a distinctive step-like feature is evident on its Pareto frontier around a watermark detection performance level of $A \approx 0.5$. This track comprised images watermarked with either GaussianShading or StegaStamp. The observed step likely indicates that one of these methods

---

[1] https://github.com/erasinginvisible/eval-program
[2] https://github.com/erasinginvisible/worker-container
[3] https://huggingface.co/datasets/furonghuang-lab/ETI_Competition_Data

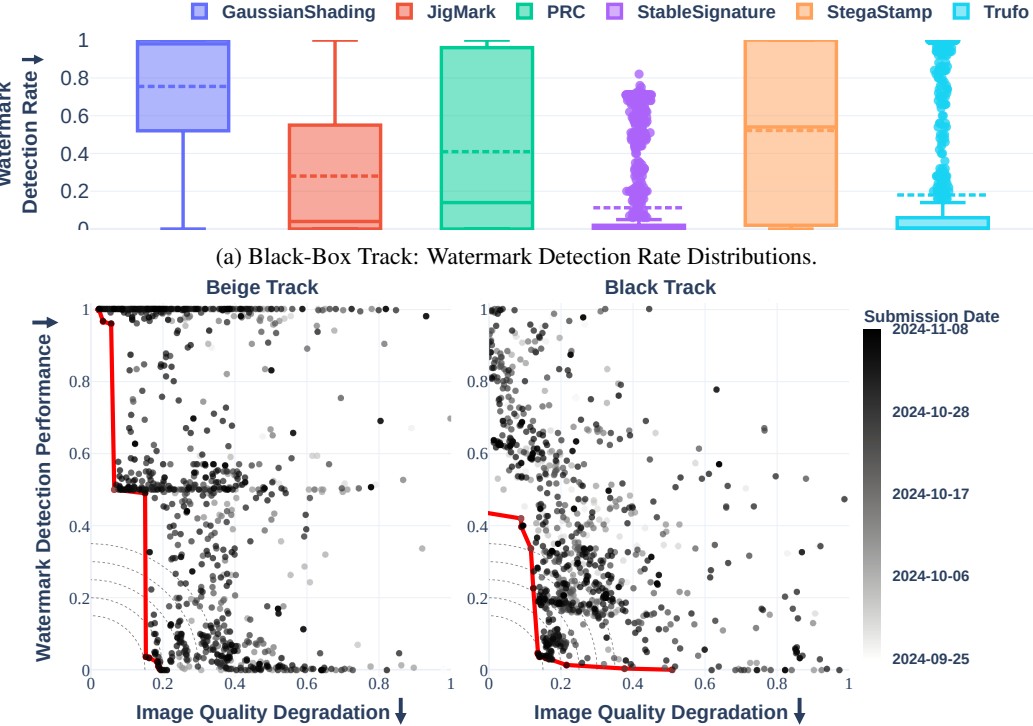

(a) Black-Box Track: Watermark Detection Rate Distributions.

(b) Submission Performance: Beige-Box and Black-Box Pareto Frontiers.

Figure 2: Overview of competition submission statistics. (a) Detection rate distributions for six Black-box track watermarks (GaussianShading, JigMark, PRC, StableSignature, StegaStamp, Trufo), revealing their relative resilience to participant attacks. (b) Scatter plots illustrating the trade-off between watermark detection performance ($A$, lower is better for attackers, indicating successful removal) and image quality degradation ($Q$, lower is better) for Beige-box (left) and Black-box (right) track submissions, with achieved Pareto frontiers highlighted in red.

(e.g., StegaStamp) was comparatively easier for many participants to break. Thus, submissions successfully attacking only this more vulnerable watermark type, thereby significantly lowering its detection performance, would cluster around $A \approx 0.5$, while the detection performance for the other, more robust watermark remained high.

In the Black-box track, the Pareto frontier appears smoother. The distribution of submissions, particularly those forming the frontier, clearly demonstrates that optimal solutions necessitate a balance between minimizing residual watermark detection performance and minimizing visual distortion. Submissions excelling in one metric at the extreme expense of the other are generally suboptimal. This outcome validates the design of our overall scoring metric (Euclidean distance to the origin in the $Q - A$ space), as it effectively encouraged participants to develop attacks that simultaneously achieve low watermark detection performance (high removal) and high image fidelity (low degradation), even under the rigorous stress-testing conditions of the competition.

## 3.2 Advantages of Watermark Superposition

A key exploration within the competition's black-box track was the efficacy of watermark superposition as a strategy to enhance robustness. The underlying hypothesis is that combining two distinct watermarking techniques onto a single image could create a more resilient defense, as an attacker would need to successfully neutralize both embedded signals. This approach is conceptually straightforward and potentially offers a practical, low-overhead method for bolstering watermark security against increasingly sophisticated removal attacks. Our findings indicate that this strategy indeed holds considerable promise, particularly when the combined watermarks leverage diverse operational principles.

**Superposition as a Layered Defense Strategy.** In our black-box track, superposition was implemented by independently injecting two different watermarks (referred to as Watermark A and Watermark B) into each test image. During the evaluation, an image was considered to still possess a valid watermark if *either* Watermark A *or* Watermark B was successfully detected by their respective decoders after an attack. Consequently, for an attack to be deemed successful in removing the watermark from a superposed image, the attacker must simultaneously eradicate or disable *both* Watermark A *and* Watermark B. This requirement inherently increases the complexity and effort needed for successful watermark removal.

**Evaluating Superposition: Setup and Metrics.** We tested two specific pairs of superposed watermarks: (1) GaussianShading combined with JigMark, and (2) StableSignature combined with StegaStamp. Table 1 summarizes the average percentage of watermarks removed by the top-5 participating teams. This metric represents the attack success rate, where a lower percentage indicates higher robustness of the watermarking scheme. The table is organized to facilitate comparison between individual watermarks and their superposed versions.

Table 1: Effectiveness of watermark superposition in the black-box track. The table shows the average percentage of watermarks removed by top-5 teams (lower is better). We tested two primary superposition pairs: (GaussianShading + JigMark) and (Stable Signatur + StegaStamp).

| Watermark Method | GaussianShading | JigMark | GaussianShading + JigMark | Stable Signature | StegaStamp | Stable Signature + StegaStamp |
|---|---|---|---|---|---|---|
| **% Watermark Removed** | 97.5% | 98.4% | **61.2%** | 100.0% | 90.2% | **93.6%** |

**Significant Robustness Gains with Dissimilar Pairings.** The results in Table 1 strikingly demonstrate the power of superposition, particularly with the GaussianShading + JigMark pairing. While GaussianShading and JigMark individually exhibited high removal rates of 97.5% and 98.4% respectively, their superposition drastically reduced the removal rate to just 61.2%. This remarkable improvement—an absolute reduction in successful attacks by over 36% compared to their individual vulnerabilities—highlights a strong synergistic effect. It suggests that combining watermarks leveraging fundamentally different operational principles, as these two methods arguably do, can significantly elevate the difficulty of successful removal by forcing attackers to overcome multiple, varied defense layers.

## 3.3 Modern Watermarks are Far from Robust

From Section 3.1, we can see that the state-of-the-art watermarks are far from robustness. The winning teams are able to remove a large part of the watermarks. Table 2 summarizes the vulnerabilities of each watermark identified by the winning teams. Note that StegaStamp and GaussianShading are tested in the beige-box setting where the attacks are more tailored, and others are tested in the black-box setting where attacks are more general.

**Most effective attacks.** The most effective attacks include geometric manipulation such as shifting and cropping, image regeneration through off-the-shelf models or specifically trained models, and watermark overwriting utilizing the white-box watermark encoder and decoders. Among them, the regeneration attack is the most widely applied and effective. Although many participants explored similar strategies, success was often hidden in subtle implementation details. For example, two teams independently identified 7 or 8 as the "magic number": shifting the image by exactly 7 or 8 pixels produces the strongest attack on GaussianShading while preserving quality. In the black-box track, many teams first group images according to their watermark patterns, whether in the spatial or frequency domain, and then tailor distinct attack strategies to each cluster.

**Strategies to maintain quality.** The main challenge in watermark removal is preserving visual quality. To address this, nearly every team introduced bespoke techniques to minimize degradation while erasing the watermark. For example, some teams applied a pixel shift and then employed a diffusion model to in-paint the resulting edge artifacts; one team dynamically adjusted their attack strength based on each image's entropy; and yet another team fine-tuned a variational autoencoder (VAE) specifically to reduce quality loss.

We defer details of the winning solutions to the Appendix but highlight the insights in Table 2.

Table 2: Vulnerabilities of tested watermarks identified by winning attacks.

| | Shiftting | Cropping & Resizing | Regeneration via Pre-trained Models | Regeneration via Custom-Trained Models | Watermark Overwrite |
|---|---|---|---|---|---|
| **StegaStamp** | — | Resize-crop distortion with dynamically adjusted cropping parameters. | Rinsing regeneration with entropy-based strengths, addition of average watermark pattern. | Fine-tune a VAE with paired datasets for watermark removel. Apply additional test-time optimization and color-contrast transfer. | 1. Utilize the StegaStamp model to overwrite watermark with repeated random message. 2. Utilize the StegaStamp model to extract message and re-embed the inversed message. |
| **GaussianShading** | Horizontal/vertical shift by 7 or 8 pixels, and inpaint the edge with a diffusion model. | Resize-crop distortion with dynamically adjusted cropping parameters. | Rinsing regeneration with FLUX-dev. | — | — |
| **StableSignature** | — | — | 1. Regeneration via a stable diffusion or FLUX model. 2. Rinsing regeneration via FLUX.1-dev and ControlNet Canny for edge detection and maintenance. Adjust attack strength based on entropy. Enhance the quality by PairOptimizer. | Controllable regeneration with semantic control and spatial control. | — |
| **PRC** | — | — | Same as above. | Controllable regeneration with semantic control and spatial control. | — |
| **Trufo** | — | — | Same as above. | 1. Fine-tune a VAE with paired datasets for watermark removel. 2. Controllable regeneration with semantic control and spatial control. | — |
| **JigMark** | — | — | Same as above. | Controllable regeneration with semantic control and spatial control. | — |
| **GaussianShading + JigMark** | — | 0.98 Cropping combined with a 3-degree rotation. | Same as above. | Controllable regeneration with semantic control and spatial control. | — |
| **StableSignature + StegaStamp** | — | — | Same as above. | 1. Fine-tune a VAE with paired datasets for watermark removel. 2. Controllable regeneration with semantic control and spatial control. | — |

### 3.3.1 Insights from the Winning Beige-Box Attacks

In the beige-box track, the participants tailored their attacks for each watermark.

**StegaStamp is vulnerable to tailored attacks.** Teams predominantly leveraged the open sourced StegaStamp encoder and decoder to design targeted approaches. The first place team generated a custom dataset using images processed with StegaStamp and their inverted messages to fine-tune a VAE that removes invisible watermarks by minimizing MSE loss between images with opposite messages [Shamshad et al., 2025]. The second place team confused the StegaStamp decoder by encoding random messages multiple times with varying perturbation strengths into the target images, effectively overwriting the original watermark. The fourth place team also performed a watermark overwriting attack by extracting, inverting, and re-embedding the hidden messages using StegaStamp's encoder and decoder networks [Serzhenko et al., 2025]. The third and fifth place teams found that regeneration attacks [Jafari, 2024] and resize-crop distortions also work on StegaStamp.

**GaussianShading, a TreeRing variant, is vulnerable to simple geometric manipulations.** For the TreeRing watermark variant, the GaussianShading, we are surprised that many successful attacks only used simple geometric manipulations. The 1st and the 2nd teams both use shifting, and they found shifting by 7-8 pixels has the best attack-quality trade-off. Due to the special design of Tree-Ring, they identified a significant vulnerability of it to phase attacks in the frequency domain, which equals to the shifting in the spatial domain. The 3rd and 5th team applied cropping and rescaling distortion method. Besides, the 4th team also found rinsing regeneration attack effective.

These diverse strategies underline the vulnerability of current watermarking techniques to both targeted attacks and surprisingly simple geometric distortions. We see innovative attacks include dataset-driven VAE training, message overwriting through perturbations, entropy-guided diffusion model adjustments, and straightforward yet highly effective geometric manipulations.

### 3.3.2 Insights from the Winning Black-Box Attacks

In the black-box track, participants designed hybrid attacks and made use of image clustering by visual artififacts.

**Watermarks Leave Distinct Visual Artifacts.** Several teams clustered images into groups based on spatial and frequency domain artifacts. Then, they would apply tailored attacks according to each cluster. This suggests that watermarks which are known to produce consistent image alterations are easier to identify and remove. For example, teams who could identify that a Tree-Ring watermark was likely present utilized pixel shifts and/or cropping which found success in the beige-box track.

**GaussianShading + JigMark is a resilient watermark.** Images using compositions of Gaussian-Shading and JigMark were strikingly resilient. Teams 2, 4, and 5, who achieved >90% removal for all other methods, struggled to remove this type of watermark. Notice that in the beige-box track teams were very successful at removing GaussianShading, while in the black-box track all JigMark images were successfully attacks. This suggests that watermarks may synergistically improve each other's integrity.

## 4 Discussions

### 4.1 Lessons Learned

Based on the most successful attacks, we describe some of the most obvious vulnerabilities of watermarks and how they may be exploited. While the setup of this competition was "red teaming" in nature, our hope is that this analysis may help inform stronger defenses and robust watermark design.

**Beware of "Simple" Geometric Attacks.** In-processing watermarks, or those which construct watermarks as part of the image generation, such as Tree-Ring [Wen et al., 2023] and Gaussian Shading [Yang et al., 2024] are considered particularly resilient [An et al., 2024, Zhao et al., 2024]. However, teams demonstrated that even simple geometric manipulations, such as pixel shifting, cropping, resizing and rotation can effectively destroy these watermarks. The creators of the Gaussian Shading were aware of the potency of phase attacks on Tree-Ring in the frequency domain, which is equivalent to spatial shifting, and subsequently inherited this vulnerability. Although naive a shift or crop would generally be detectable by the human eye (by leftover black columns/rows), circular shifting and in-painting to fill in disturbed pixels can eliminate these artifacts.

**Post-processing Boosts Regenerative Attacks.** Several teams used fine-tuned VAEs, Stable Diffusion Refiner models, and ControlNets to disturb the latent structure of images in an attempt to erase the watermark. Despite their best attempts at hyperparameter tuning, additional post-processing was still required to further restore image quality. The winning team of both the beige-box and black-box tracks used optimized color-contrast restoration in CIELAB space [Shamshad et al., 2025]. The black-box second place team and beige-box third place team [Jafari, 2024] developed their own customized tool to fine-tune color properties (exposure, gamma, brightness, hue, tint, etc.) to improve PSNR and MS-SSIM [Wang et al., 2003] loss.

**Vanilla Bases of Custom Models Can Defeat Beige Boxes.** Our goal in revealing the general watermarking type for images in the beige-box track was to assess how effectively attackers could work in the absence of the watermarking model itself. Despite the usage of custom StegaStamp and Gaussian Shading models for our images, off-the-shelf versions of these models could be successfully fine-tuned to transfer attacks. For the StegaStamp images, the fifth place team of the beige-box track synthetically encoded images with random binary messages via a vanilla encoder and then fine-tuned a vanilla decoder to invert the messages [Serzhenko et al., 2025]. The winner of the beige-box track improved upon this approach by fine-tuning a VAE and CIELAB color transfer to bring these images closer together in visual quality.

**Combining Watermarks Can Improve Robustness.** Winning teams on the black-box track achieved an average >90% attack success rate on all watermarking types with the lone exception of Gaussian Shading + JigMark (see Table 6). The winning teams could only remove 76.5% of this type of watermark on average, excluding an outlier team that could not remove it from *any of the images*. As detailed in Section 3.3.2, these watermarks were individually easy to defeat, thus their combination appears to emergently produce a stronger watermark.

## 4.2 Open Questions

Organizing this competition motivated us to think deeper about the future of watermarks. Despite significant progress, the field faces a number of unresolved challenges that call for further exploration. Below are some of the many challenges.

**Detectability in Open-Source Models.** As open-source generative models reach the sophistication of proprietary "black-box" systems, how can we reliably embed and detect invisible watermarks in their outputs—especially when there is no unified interface and community-driven forks?

**Verifiability of "Non-Generation" Claims.** If a given model asserts that it did not generate an image, by what means can we demonstrate that no other model (proprietary or open-source) produced it? Excluding every known watermark fingerprint represents an almost intractable, large-scale challenge.

**Feasibility of a Universal Invisible Watermark Scheme.** Given the diversity of existing algorithms, is it possible to design a single, cross-model standard watermark that remains imperceptible, robust against attacks, and easy to integrate?

**Tamper-Resistance and Security.** Open-access detection APIs facilitate broader research and deployment but also empower adversaries to reverse-engineer and strip watermarks (e.g., via noise injection or geometric transforms). How can embedding mechanisms (e.g., stronger encryption, randomized seeding) and detection pipelines (e.g., robust machine-learning classifiers) be co-designed to resist such attacks end-to-end?

## 5 Conclusion

Watermarks are a useful tool for establishing image provenance. We organized the NeurIPS 2024 "Erasing the Invisible" competition to determine the robustness of many popular watermarking algorithms. We created a beige-box and black-box track to determine how attack strategies would evolve in the presence of information (or lack thereof). In fact, teams were able to successfully remove most watermarks while minimally disturbing image quality. This necessitates further research into stronger watermarks and defenses. One potential path forward is to study combinations of watermarks, which our competition has shown to improve robustness.

## Ethical Statement

Our work proceeds from the principle that resilient defenses emerge from a clear understanding of plausible attacks. We detail vulnerabilities entirely in the spirit of responsible disclosure to catalyze a more robust generation of watermarking technology. By focusing on open-source watermarking algorithms, we can inform the academic community about weaknesses without directly threatening systems using watermarks. We intentionally do not release attack implementations. We remind the reader that our results contain strong defensive takeaways, in particular by demonstrating that combining dissimilar watermarks offers a powerful defense. We also note that simple geometric attacks, while effective, can leave traces detectable by forgery-detection networks. We believe that our released dataset is a resource for developing such tools.

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

# Appendices

## A    Limitations

As described in Sections 4 and 6, users would tailor attacks to image clusters. In the case of beige box, we outright provided these clusters by disclosing which image indices corresponded to which general watermark type. For the black-box track, several winning teams clustered images into groups by artifact varieties and did so by hand. For the latter, this was made possible because (1) our data set was relatively small, enabling this type of manual data labeling, and (2) they were made aware that the dataset contained mixtures of several watermarks. A database owner who uses only one type of watermark will unlikely produce such variation in artifacts.

Additionally, we use the watermark models and setting provided in the original papers and do not calibrate the strength of watermarks. Therefore, the comparison of watermarks' robustness could be biased. For example, images watermarked by StegaStamp shown visible artifacts that hurt the image quality and provide clues of the watermark used. Calibration watermarks is challenging since different watermarks use different strategies. One promising solution that future work could consider is adjusting the strength of watermarks (e.g., message length) so that the quality degradation of watermarked images are the same.

## B    Broader Impact

Erasing the Invisible brings together a global community to rigorously evaluate the resilience of invisible watermarks in AI-generated images, uncovering critical vulnerabilities in methods once deemed robust. These findings will directly inform the design of next-generation watermarking schemes, helping content creators, platforms, and policymakers deploy more reliable provenance tools to combat misinformation, copyright infringement, and evidence tampering.

Moreover, the competition pipeline can be effortlessly extended to live or rolling benchmarks, enabling continual evaluation of emerging watermarking techniques. By providing an open, standardized benchmark, we enable reproducible progress in both attack and defense, ultimately strengthening trust in digital media.

## C    Acknowledgment

**Technical Support**

We extend our sincere thanks to the authors of the watermarking methods used in this competition. Their permission to employ their models and assistance in setting them up made this work possible. Special thanks to Xuandong Zhao and Sam Gunn for providing PRC before publication, Minzhou Pan and Yi Zeng for training a JigMark model specifically for this competition, and Trufo for providing their API.

Although certain contributions were ultimately not adopted for various reasons, we are deeply grateful to everyone who supported us along the way. Thanks to Ashley Chow and Ryan Holbrook from Kaggle for their effort in setting up the infrastructure for us at the initial stage. Thanks to Vikash Sehwag for providing a secret diffusion model.

**Sponsors**

A massive shoutout to the UMIACS computing facilities team, led by Derek Yarnell, who worked tirelessly with us 24/7 to keep the servers running smoothly throughout the competition. Their technical support was absolutely vital, ensuring we could handle the large volume of submissions efficiently. We also want to express our sincere gratitude to Emily Hartz, Executive Director of Administration & Operations, and Petra Zapf, Director of Finance, for helping us navigate the legal complexities surrounding our prize distribution. Their commitment to the success of this competition was unmatched. And, of course, none of this would have been possible without Tom Goldstein, Director of the Center for Machine Learning. Tom provided invaluable financial and technical support. He helped us tackle challenges head-on, all while keeping the spirit of innovation alive. His leadership

and the entire team's effort turned this competition from a concept into reality. We're incredibly grateful for the collaborative energy and support from UMIACS and the Center for Machine Learning.

# D    Related Work

## D.1    Benchmarking Attacks.

The authors of a new watermark will typically demonstrate their robustness by subjecting them to a large number of attacks. A survey of few modern methods [Fernandez et al., 2023, Tancik et al., 2020, Wen et al., 2023, Pan et al., 2024, Yang et al., 2024] reveals that they were benchmarked over differing datasets, attack types (and intensities), and $p$-values for attack rejection (i.e., the threshold for not accepting a watermark was removed). Attack authors similarly did not assess the same watermark types [Nie et al., 2022, Saberi et al., 2024] and/or knowledge scenarios [Lukas et al., 2023, Jiang et al., 2023]. This spurred the creation of this competition [Ding et al., 2024], to catalog a greater collection of user-submitted attacks according to the principles of a standardized robustness benchmark, WAVES [An et al., 2024]. Although winners had to disclose their attack algorithms, with several already publicly available as pre-prints or notes [Shamshad et al., 2025, Serzhenko et al., 2025, Jafari, 2024], the general user was not required to describe any submitted attack. A pseudo-anonymous, publicly-available leaderboard of attacks is novel.[4]

## D.2    Modern Watermarks

Watermark design is an active area of research. We refer the reader to [Zhao et al., 2024, Fernandez et al., 2023, Gunn et al., 2023, An et al., 2024] for surveys of modern generative watermarks. For our competition, we selected watermarks of in-processing and post-processing types, (following the taxonomy of [Ding et al., 2024, An et al., 2024]).

For post-processing watermarks, we used *(1)* the StegaStamp [Tancik et al., 2020], a watermark designed for preventing photographic theft, with enhanced robustness via attack-discrete adversarial training *(2)* the JigMark [Pan et al., 2024] which resists image editing by using an encoder which learns to embed a watermark in Fourier low-frequency bands. *(3)* an industry watermark developed by Trufo. It is a Y-channel watermark which targets the noisier regions of images. The exact method is proprietary.

For in-processing watermarks, we used *(1)* the Stable Signature [Fernandez et al., 2023], which trains the decoder module of a Stable Diffusion pipeline to embed a message. *(2)* Gaussian shading [Yang et al., 2024] embeds a message into the latent representation of image which follows a Gaussian distribution, thus preserving the latent space. *(3)* PRC [Gunn et al., 2023], which embeds a cryptographically pseudorandom pattern into the latent space and can be decoded via an error-correcting code.

# E    Detailed Recap of the Competition

## E.1    Competition Structure and Design

The competition was structured into two distinct tracks designed to probe watermark robustness under different attacker knowledge scenarios: the **Beige-box Track** and the **Black-box Track**. This dual-track approach allowed for a nuanced evaluation, assessing resilience both when attackers have partial information and when they operate with minimal knowledge, simulating diverse real-world conditions.

A fundamental distinction in watermarking techniques lies between *in-processing* methods, which embed the watermark during the generative process (typically within AI models), and *post-processing* methods, which apply the watermark to an already generated image (whether AI-generated, natural, or human-crafted). Our competition incorporated examples of both paradigms.

The images used across both tracks were generated using a hybrid approach, employing both the **Flux. 1 [dev]** model and **Stable Diffusion 2.1**. Prompts were designed to elicit diverse content and

---

[4]OpenAI developed their own Detection Classifier Program, which was a whitelist of testers to attack and challenge the detectability of DALL-E-generated content. However, these results are currently not available.

styles, ensuring a broad testbed for the watermarking techniques. A critical aspect of the competition design was the alignment of watermark strength. Prior to the competition, we meticulously tuned the hyperparameters for each deployed watermark method to achieve a standardized baseline robustness, targeting a **False-Positive Rate (FPR) of 0.1%**. This ensured that, without any attack, approximately 99.9% of watermarked images could be correctly detected, providing a consistent starting point for evaluating removal efficacy.

### E.1.1  Beige-box Track Details

In the Beige-box track, participants were provided with the names of the watermarking algorithms applied, enabling more targeted attacks and testing the fundamental limits of these specific techniques. We employed two representative methods, distributed across 300 images:

- **Gaussian Shading** Yang et al. [2024]: An *in-processing* watermark applied to 150 images generated by Stable Diffusion 2.1. This method embeds a unique pattern within the initial noise vector used by the diffusion model during image synthesis. Detection involves reversing the diffusion process to recover the noise vector and check for the embedded pattern.
- **StegaStamp** Tancik et al. [2020]: A *post-processing* watermark applied to 150 images generated by Flux. 1 [dev]. StegaStamp utilizes a deep neural network trained to create robust, invisible watermarks via learned encoding and decoding mechanisms, designed to withstand various distortions, including print-scan operations.

Providing the method identity allowed participants to potentially leverage public knowledge or implementations to devise sophisticated removal strategies.

### E.1.2  Black-box Track Details

The Black-box track aimed to simulate a more realistic adversarial scenario where the attacker possesses limited or no prior knowledge about the specific watermarking technology employed. This track featured a total of 300 images, comprising a more complex mix of watermarks kept entirely confidential from the participants:

- **Singly-Watermarked Images (200 total)**: 50 images each were watermarked using one of four distinct methods: JIGMARK Pan et al. [2024], PRC Gunn et al. [2023], Stable Signature Fernandez et al. [2023], and the Trufo watermark.
- **Doubly-Watermarked Images (100 total)**: To increase the challenge and reflect potential real-world practices, we included images watermarked with combinations: 50 images with Gaussian Shading + JIGMARK, and 50 images with Stable Signature + StegaStamp.

All images in this track were shuffled, preventing participants from inferring methods based on order or batching. We intentionally maintained secrecy around the methods used in this track because real-world watermarking deployments often rely on confidentiality, alongside other engineering enhancements like proprietary implementations, randomized keys, multi-watermark layering, and potential (though not explicitly tested here) anti-tampering defenses. Evaluating robustness under these conditions provides a more practical assessment of watermark resilience against uninformed attacks.

### E.2  Evaluation Metrics and Scoring

The competition aimed to rigorously assess the trade-off between watermark removal efficacy and the preservation of image quality. To quantify this, we developed a sophisticated, automated evaluation system based on the principles established in the WAVES benchmark An et al. [2024]. Each submission was assigned a final score calculated as the Euclidean distance in a 2D performance-quality space:

$$\text{Score} = \sqrt{Q^2 + A^2}$$

where $Q$ represents the normalized image quality degradation and $A$ represents the watermark detection performance (specifically, $1 - \text{TPR@0.1\%FPR}$). The primary goal for participants (attackers) was to **minimize this score**, signifying successful watermark removal ($A \approx 1$) with minimal impact on image quality ($Q \approx 0$).

### E.2.1 Performance Metric

Watermark detection performance was measured by the True Positive Rate (TPR) at a stringent False Positive Rate (FPR) of 0.1% (TPR@0.1%FPR). This metric reflects the ability to correctly identify watermarked images while maintaining an extremely low rate of misclassifying unwatermarked images.

- **Threshold Determination:** For each watermarking algorithm used in the competition, we first established a detection threshold. This involved running the specific watermark decoder on a large set of 10,000 diverse, unwatermarked images (generated from the same prompt distribution as the competition data). The distribution of detection scores (e.g., message distance $d(m, m')$ from the expected message $m$) from these unwatermarked images was analyzed, and the score corresponding to the 0.1 percentile (i.e., the value below which only 0.1

- **TPR Calculation:** During evaluation, participants' attacked images were processed using the relevant watermark decoders. An image was considered "detected" if its decoding score fell below the pre-calculated 0.1

- **Final Metric A:** The value used in the scoring formula is $A = 1 - \text{TPR@0.1\%FPR}$, representing the watermark removal rate. A value of $A = 1$ indicates complete removal (TPR=0), while $A = 0$ indicates no removal (TPR=1).

### E.2.2 Normalized Image Quality Degradation

To comprehensively assess the visual impact of the attacks, we computed a normalized image quality degradation score, $Q$. This score aggregates information from eight distinct image quality metrics (IQMs), comparing the attacked image to its original, unattacked (but potentially watermarked) version: (i) Peak Signal-to-Noise Ratio (PSNR) (ii) Structural Similarity Index (SSIM) Wang et al. [2004] (iii) Normalized Mutual Information (NMI) (iv) Frechet Inception Distance (FID) Heusel et al. [2017] (v) CLIP Image Fidelity (CLIP-FID) Kynkäänniemi et al. [2019] (vi) Learned Perceptual Image Patch Similarity (LPIPS) Zhang et al. [2018] (vii) Delta Aesthetics Score ($\Delta$Aesthetics) Xu et al. [2023] (viii) Delta Artifacts Score ($\Delta$Artifacts) Xu et al. [2023] The normalization procedure, detailed in An et al. [2024], involved:

- Establishing baseline distributions for each IQM by applying a diverse set of 26+ known attacks to a large image corpus.
- Determining the 10th and 90th percentile scores for each metric within this corpus, representing low and high degradation levels, respectively.
- Normalizing the IQM score for each submitted attacked image to the range [0.1, 0.9] based on these percentiles (scores outside this range were clamped).
- Calculating the final $Q$ score as a weighted average of these normalized IQM scores, using empirically derived coefficients:

$$Q = +1.53 \times 10^{-3} \, \text{FID} + 5.07 \times 10^{-3} \, \text{CLIP FID} - 2.22 \times 10^{-3} \, \text{PSNR}$$
$$- 1.13 \times 10^{-1} \, \text{SSIM} - 9.88 \times 10^{-2} \text{NMI} + 3.41 \times 10^{-1} \, \text{LPIPS}$$
$$+ 4.50 \times 10^{-2} \Delta\text{Aesthetics} - 1.44 \times 10^{-1} \Delta\text{Artifacts}$$

A higher $Q$ value indicates greater image degradation (poorer quality relative to the original).

### E.3 Competition Platform and Infrastructure

The competition was hosted on the Codabench platform Farragi et al. [2020–], an open-source system for computational challenges, utilizing dedicated instances for the Beige-box [5] and Black-box [6] tracks.

To handle the computationally intensive evaluation process involving deep learning models and numerous metrics, we deployed custom compute workers. These workers were built upon the

---

[5]https://www.codabench.org/competitions/3821/
[6]https://www.codabench.org/competitions/3857/

standard Codabench worker architecture but packaged within Docker containers equipped with GPU support via the NVIDIA Container Toolkit. This ensured reproducible environments with necessary libraries (PyTorch, ONNXRuntime-GPU, Transformers, Diffusers, etc.) and allowed for parallel processing across multiple GPU devices, managed via Docker Compose and coordinated through a Celery message queue connected to the Codabench backend.

The core evaluation logic was implemented in a dedicated open-source Python package[7], executed by the compute workers. Upon receiving a submission (consisting of 300 attacked PNG images), the evaluation pipeline performed the following steps automatically:

1. **Input Verification:** Checked submission format compliance.
2. **Standardized Preprocessing:** Applied minor, standardized image manipulations (3x3 median blur, JPEG compression at QF=98) to simulate common distribution conditions.
3. **Watermark Decoding:** Executed the relevant decoding algorithms for the track (Beige-box known methods or Black-box secret methods).
4. **Quality Assessment:** Computed the eight IQMs described in the Evaluation Metrics section by comparing the preprocessed submission to pristine reference images. Required models for metrics like LPIPS and CLIP-FID were dynamically fetched from the Hugging Face Hub.
5. **Scoring & Output:** Calculated the performance metric $A$ and quality metric $Q$, computed the final score $\sqrt{Q^2 + A^2}$, and reported results back to Codabench.

This automated backend enabled a **real-time rolling leaderboard**, providing participants with immediate feedback on their submission's performance and ranking. To complement the automated metrics and ensure fairness, especially in cases of close scores or potential metric exploitation, the top-ranked submissions in each track underwent an additional layer of **human evaluation** by the organizers, focusing on subjective visual quality assessment.

## F    Competition Submission Statistics and Activity

The "Erasing the Invisible" competition, hosted on the Codabench platform[8], ran from September 16 to November 5, 2024. It attracted significant global engagement, with a total of 2,722 submissions received from 298 participating teams worldwide, underscoring the community's strong interest in evaluating and advancing image watermark robustness. The Beige-box track saw 1,072 submissions from 65 distinct teams, while the Black-box track recorded 1,650 submissions from 77 distinct teams.

The competition's progression and outcomes are further illustrated by the following figures. Figure 3 provides a comparative look at the final score distributions for both tracks, highlighting the range and concentration of participant performance. Figure 4 details the engagement dynamics, showcasing the daily and cumulative submission counts throughout the competition period, reflecting bursts of activity and sustained effort from the participants. Finally, Figure 5 visualizes the evolution of the best-achieved scores over time, demonstrating the competitive landscape and the gradual improvement in attack efficacy as teams refined their strategies. These statistics collectively depict a highly active and competitive challenge.

## G    Public Dataset Release

To foster continued research and transparency, all data generated from the "Erasing the Invisible" competition has been publicly released on Hugging Face under the dataset ID `furonghuang-lab/ETI_Competition_Data`[9]. This comprehensive dataset is licensed under Creative Commons Attribution 4.0 International (CC BY 4.0) and serves as a valuable resource for researchers in digital watermarking, adversarial machine learning, and content authenticity.

The dataset is structured into four primary subsets:

---

[7]`https://github.com/erasinginvisible/eval-program`

[8]Beige-box track: `https://www.codabench.org/competitions/3821/`, Black-box track: `https://www.codabench.org/competitions/3857/`

[9]`https://huggingface.co/datasets/furonghuang-lab/ETI_Competition_Data`

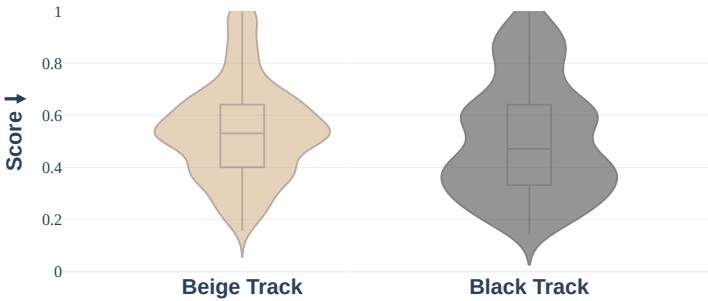

Figure 3: Final score distributions for the Beige-box and Black-box tracks. The violin plots illustrate the density of participant scores (lower is better, Score = $\sqrt{Q^2 + A^2}$), including median and interquartile ranges, providing insight into overall performance and score clustering within each track.

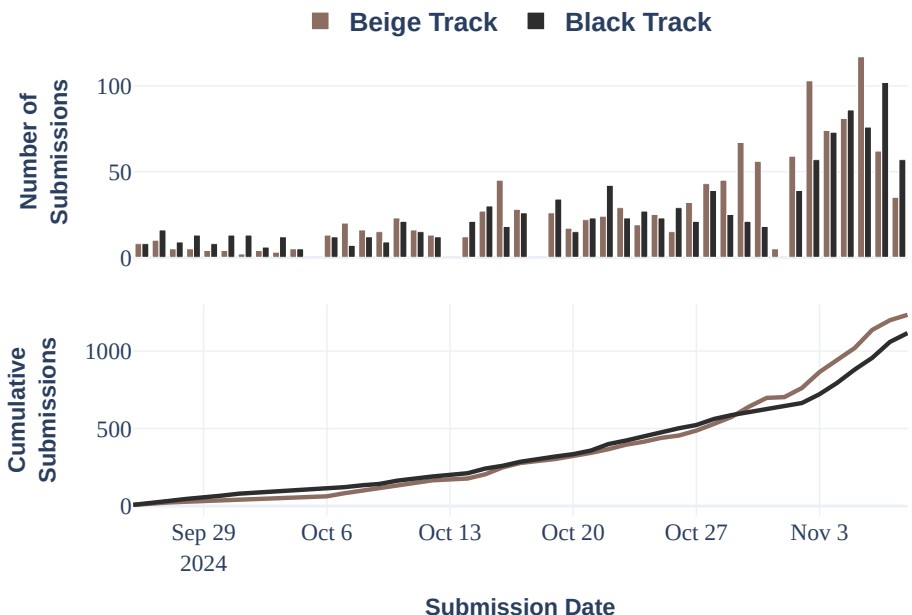

Figure 4: Submission activity throughout the competition (September 29, 2024 - November 10, 2024, as shown in figure). The top panel displays the number of daily submissions for both Beige-box (brown) and Black-box (black) tracks, indicating periods of heightened activity. The bottom panel shows the cumulative number of submissions over time for each track, illustrating the overall engagement.

- `Beige_Track_Images`: Contains the 300 original images used in the Beige-box track, watermarked with either Gaussian Shading (150 images from Stable Diffusion 2.1) or StegaStamp (150 images from Flux.1 [dev]). Each entry includes the `image_index` and the `watermarked_image`.

- `Black_Track_Images`: Contains the 300 original images for the Black-box track, featuring a confidential mix of watermarks. This includes 50 images each for single watermarks (Jig-Mark, PRC, StableSignature, Trufo) and 50 images each for double watermarks (Gaussian Shading + JigMark, StableSignature + StegaStamp). Each entry includes the `image_index` and the `watermarked_image`.

- `Beige_Track_Submissions`: Provides detailed evaluation metadata and scores for all 1,072 valid submissions to the Beige-box track. Key features include `submission_id`, `submission_time`, dictionaries with per-watermark (`gaussianshading`, `stegastamp`)

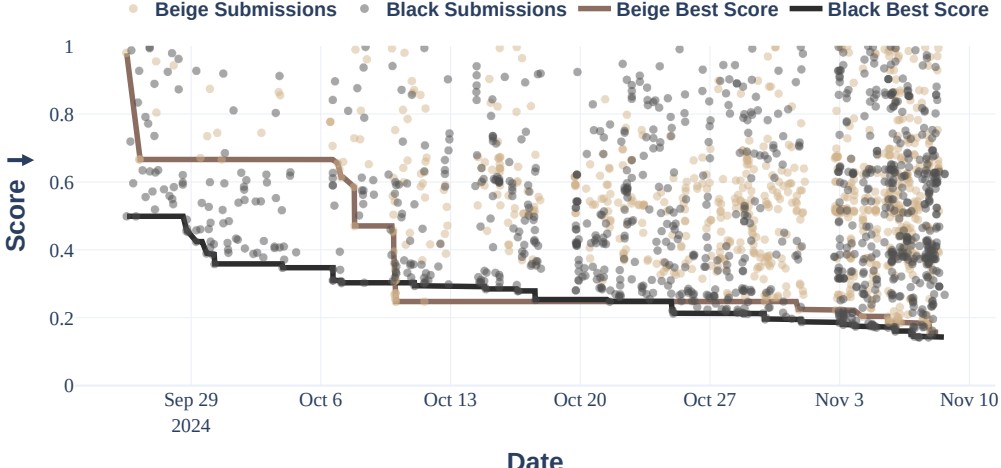

Figure 5: Evolution of submission scores over the competition period (September 29, 2024 - November 10, 2024, as shown in figure). Each point represents a submission, with beige indicating Beige-box track submissions and black indicating Black-box track submissions. The solid lines (brown for Beige-box, black for Black-box) trace the evolution of the best achieved score (Pareto frontier) over time, demonstrating continuous improvement in attack strategies. Lower scores indicate better performance.

      and per-image IQM scores (`aesthetics`, `artifacts`, `clip_fid`, `legacy_fid` (FID), `lpips`, `nmi`, `psnr`, `ssim`), and the final `performance` ($A$), `quality` ($Q$), and overall `score`.

- `Black_Track_Submissions`: Contains corresponding evaluation metadata and scores for the 1,650 valid submissions to the Black-box track. Features are similar to the Beige-box submissions, with per-watermark score dictionaries for `gaussianshading`, `jigmark`, `prc`, `stablesig`, `stegastamp`, and `trufo`.

The dataset includes not only the evaluation scores but also allows access to the actual attacked image files submitted by participants, enabling in-depth analysis of attack strategies. Users can load specific subsets or the entire dataset using the Hugging Face `datasets` library. For detailed instructions on accessing attacked images and the full schema, please refer to the dataset card on Hugging Face. This resource is intended to support the development of more robust watermarking techniques and better evaluation methodologies.

# H Open-Source Evaluation Toolkit

To ensure transparency, reproducibility, and facilitate future research, the complete evaluation infrastructure for the competition is open-sourced under the Apache License 2.0. This includes the core evaluation program and the Codabench worker container setup.

## H.1 Evaluation Program

The core evaluation logic is available on GitHub at `erasinginvisible/eval-program`[10]. This Python-based program was responsible for processing each participant submission (a set of 300 attacked images). Its functionalities include:

- **Input Verification**: Ensuring submissions adhere to the specified format.
- **Standardized Preprocessing**: Applying minor image manipulations (e.g., median blur, JPEG compression) to simulate common distribution conditions.

---

[10]`https://github.com/erasinginvisible/eval-program`

- **Watermark Decoding**: Executing the relevant watermark decoding algorithms. Separate entry points (`beige.py` and `black.py`) handle the distinct logic for Beige-box (known watermarks) and Black-box (secret watermarks) tracks.
- **Image Quality Assessment**: Computing eight distinct Image Quality Metrics (IQMs) by comparing attacked images to their original watermarked versions. Models for metrics like LPIPS and CLIP-FID were dynamically fetched.
- **Scoring and Output**: Calculating the final performance metric $A$ (watermark removal rate) and quality metric $Q$ (image degradation), combining them into the overall competition score $\sqrt{Q^2 + A^2}$, and reporting these to Codabench.

The repository includes all necessary helper functions, metric calculation scripts, and dependencies (listed in `requirements.txt`, which specifies `onnxruntime-gpu`, indicating GPU optimization). While designed for Codabench, the program can also be run locally for testing or further research.

### H.2 Codabench Worker Container

The Dockerized environment used to run the evaluation program on Codabench is available at `erasinginvisible/worker-container`[11]. This setup builds upon the standard Codabench worker architecture but is specifically configured for GPU-accelerated tasks using the NVIDIA Container Toolkit. Key aspects include:

- **Custom Docker Image**: The repository provides `Dockerfile.nvidia` to build a custom worker image (`johnding1996/codabench-erasinginvisible:latest`) equipped with necessary libraries like PyTorch, ONNXRuntime-GPU, Transformers, and Diffusers.
- **GPU Configuration**: The `docker-compose.yml` file is configured to manage multiple worker instances, allowing for parallel execution and assignment of specific GPUs to different workers.
- **Reproducible Environment**: Ensures that all submissions were evaluated in a consistent and reproducible computational environment.

These open-source tools, in conjunction with the public dataset released as described in appendix G, provide a comprehensive benchmark and a foundation for future advancements in image watermarking security and evaluation.

## I   Winners' Solutions

### I.1   Beige-Box Solutions

Table 3: Beige-box winners' scores.

| Team | Prev Overall Score [↓] | Watermark Detect Perf [↓] | Quality Degrad (Machine) [↓] | Quality Degrad (Human) [↓] | Final Score [↓] |
|---|---|---|---|---|---|
| Team-MBZUAI | 0.1570 | 0.0367 | 0.1526 | 0.1526 | 0.1570 |
| asky30 | 0.1834 | 0.0500 | 0.1764 | 0.1683 | 0.1756 |
| mohammadjafari | 0.2558 | 0.1267 | 0.2223 | 0.2221 | 0.2557 |
| hesiyang | 0.3434 | 0.0567 | 0.3387 | 0.2719 | 0.2777 |
| leiluk1 | 0.3197 | 0.1000 | 0.3036 | 0.3387 | 0.3532 |

The 1st team Shamshad et al. [2025] generated a custom dataset using images processed with StegaStamp and their inverted messages to fine-tune a VAE that removes invisible watermarks by minimizing MSE loss between images with opposite messages. They then applied post-processing techniques, including test-time VAE optimization and color and contrast transfer, to enhance image quality. Uniquely, for the TreeRing watermarked images, they discovered a vulnerability to phase attacks and effectively removed the watermark by horizontally translating images by 7 pixels, a simple yet effective method compared to other submissions.

---

[11]`https://github.com/erasinginvisible/worker-container`

The 2nd team confused the StegaStamp decoder by encoding random messages multiple times with varying perturbation strengths into the target images, effectively overwriting the original watermark. For the TreeRing watermarked images, they shifted the images 8 pixels upwards and used Stable Diffusion to inpaint the resulting blank space, disrupting the watermark. Uniquely, they combined message overwriting with varying strengths and advanced inpainting techniques to remove watermarks compared to other teams.

The 3rd team Jafari [2024] utilized a FLUX.1-dev model with ControlNet Canny for edge preservation during image manipulation. For StegaStamp images, they performed a multi-pass Img2Img pipeline with strengths adjusted based on image entropy, and added a precomputed average watermark pattern during each iteration to weaken the embedded messages. For TreeRing watermarks, they applied cropping and rescaling techniques. Uniquely, their approach included entropy-based strength adjustments and the addition of average watermark patterns, which differed from other teams' methods.

The 4th team used a resize-crop distortion method, adjusting the cropping scale dynamically based on a strength parameter to effectively remove watermarks while preserving image content. They applied different strength values for StegaStamp and TreeRing watermarked images, finding that both were vulnerable to this distortion-based attack. Uniquely, they demonstrated that even robust watermarks are susceptible to simple distortions like resize-crop, which other teams did not focus on.

The 5th team performed a Watermark Overwriting Attack Serzhenko et al. [2025], on StegaStamp images by extracting, inverting, and re-embedding the hidden messages using StegaStamp's encoder and decoder networks, effectively removing the watermark. For TreeRing images, they applied a Rinsing Regeneration Attack using multiple cycles of noising and denoising with a pretrained diffusion model (FLUX-dev). Uniquely, they combined message inversion and re-embedding with regeneration attacks to remove watermarks, which was distinct from other submissions.

Table 4: Winning teams' attacks in the Beige-box track.

| Team | Attacks on StegaStamp | | Attacks on GaussianShading | |
|---|---|---|---|---|
| | ASR | Method | ASR | Method |
| 1 | 97.3% | VAE fine-tuned with paired datasets, test-time optimization, color-contrast transfer | 95.3% | Horizontal shift by 7 pixels (phase attack vulnerability) |
| 2 | 90.0% | Overwriting watermark with repeated random message encoding at varying perturbation strengths | 100.0% | Vertical shift by 8 pixels with Stable Diffusion-based inpainting |
| 3 | 98.6% | Multi-pass Img2Img with entropy-based strengths, addition of average watermark pattern | 76.0% | Cropping and rescaling manipulations |
| 4 | 100.0% | Message extraction, inversion, and re-embedding using StegaStamp encoder-decoder | 80.0% | Rinsing regeneration with pretrained diffusion model (FLUX-dev) |
| 5 | 99.3% | Resize-crop distortion with dynamically adjusted cropping parameters | 89.3% | Resize-crop distortion with dynamically adjusted cropping parameters |

## I.2 Black-Box Solutions

The 1st team Shamshad et al. [2025] clustered the watermarked images into four groups based on observed artifacts and applied tailored methods to each. They used a Stable Diffusion Refiner Model with cluster-specific strength parameters, and for some clusters, applied their VAE-based watermark removal method, including test-time VAE optimization and color/contrast transfer. Additionally, they exploited a vulnerability by horizontally translating images by 7 pixels to perform phase manipulation.

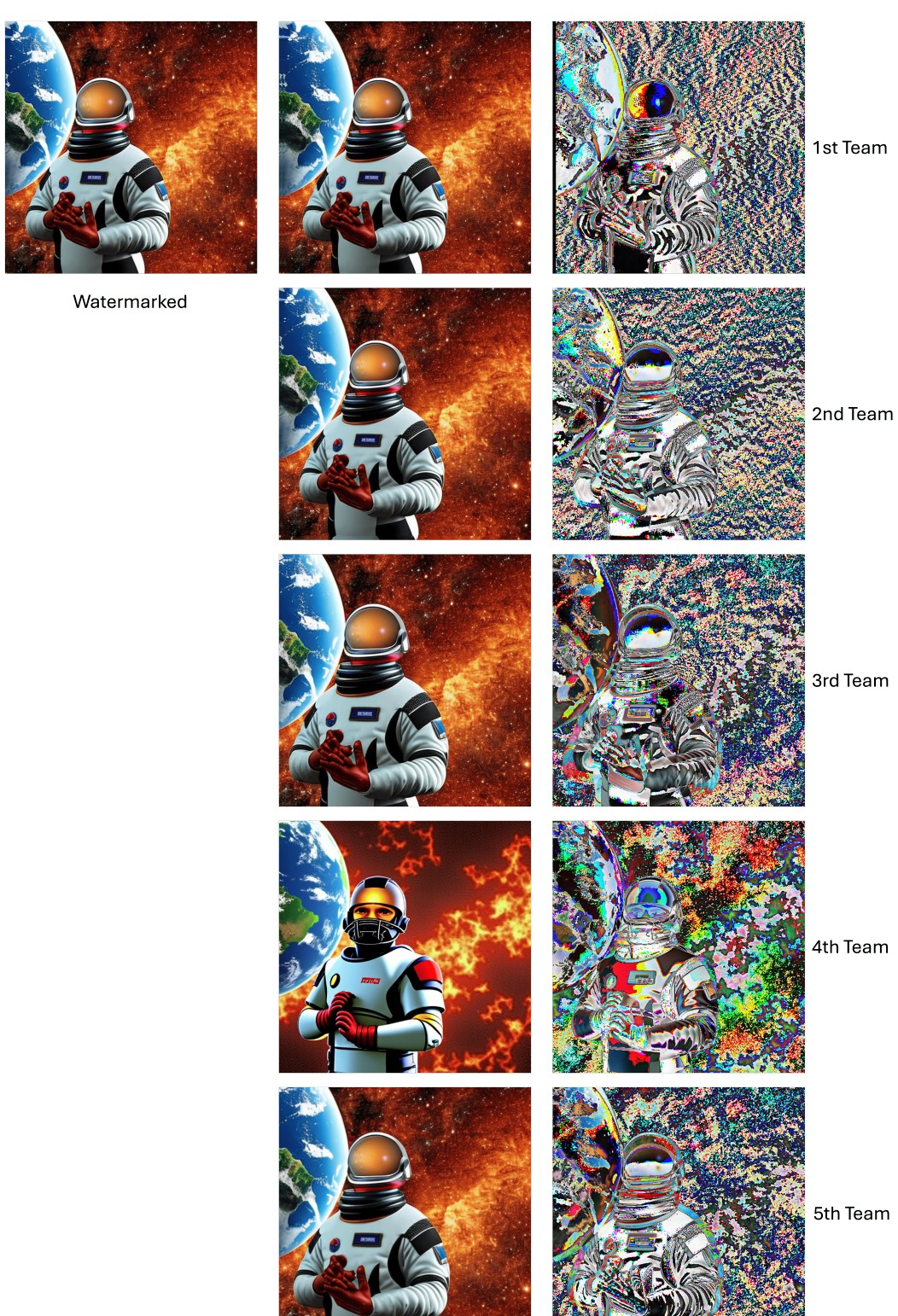

Figure 6: Examples of top 5 teams' attacks in the beige-box track.

Table 5: Black-box winners' scores.

| Team | Prev Overall Score [↓] | Watermark Detect Perf [↓] | Quality Degrad (Machine)[↓] | Quality Degrad (Human) [↓] | Final Score [↓] |
|---|---|---|---|---|---|
| Team-MBZUAI | 0.1430 | 0.0433 | 0.1363 | 0.1420 | 0.1485 |
| mohammadjafari | 0.1699 | 0.0633 | 0.1576 | 0.1363 | 0.1503 |
| asky30 | 0.2088 | 0.0667 | 0.1979 | 0.1413 | 0.1563 |
| yepengliu | 0.1973 | 0.0867 | 0.1773 | 0.1979 | 0.2161 |
| egorkov | 0.2339 | 0.1900 | 0.1365 | 0.1432 | 0.2379 |

Uniquely, their method combined clustering with multiple tailored approaches, including phase manipulation and VAE-based removal, setting them apart from other teams.

The 2nd team Jafari [2024] employed a FLUX.1-dev model with ControlNet Canny for controlled image manipulation, adjusting attack strength based on image entropy calculations to preserve quality. They maintained image structure using edge detection and resized images to improve processing. They performed purification with varying parameters and enhanced visual similarity using PairOptimizer, which fine-tunes images with differentiable adjustments. For TreeRing watermarks, they applied cropping and slight rotation. Uniquely, their method integrated entropy-based adjustments, ControlNet, and a custom post-processing tool, PairOptimizer, differing from other submissions.

The 3rd team categorized the images into two groups and for Group 1, they applied denoising using Stable Diffusion with a 'denoise: 1.0' prompt, then resized the output to the original dimensions. For Group 2, they denoised images with different prompts ('denoise', 'dehaze', 'clean'), shifted the images 7 pixels upwards, and selected the best output based on SSIM. Uniquely, they combined diffusion-based denoising with spatial shifting and optimization based on structural similarity, which was different from other teams.

The 4th team proposed Controllable Regeneration (CtrlRegen+) Liu et al. [2025], a no-box watermark removal attack that adds adjustable noise to the latent representation to disrupt watermark information. They introduced semantic control by encoding the watermarked image into an image embedding and used cross-attention mechanisms to preserve semantic content during regeneration. Additionally, they incorporated spatial control using edge-detected images to maintain structural layout via a spatial control network. Uniquely, their method combined semantic and spatial controls in a unified framework to effectively remove watermarks while preserving image quality, which was distinct from other teams.

The 5th team hypothesized that the watermark was embedded in the image's latent representation and aimed to perturb this latent vector to remove the watermark with minimal quality loss. They applied image-to-image regeneration using the FLUX model, adjusting parameters like guidance scale, noise magnitude, and inference steps to optimize results. Uniquely, they focused on perturbing the latent space via FLUX model regeneration to remove watermarks, differing from other teams' approaches.

Table 6: Winning teams' attacks in the Black-box track.

| Method Attacked | ASR | | | | |
|---|---|---|---|---|---|
| | Team 1 | Team 2 | Team 3 | Team 4 | Team 5 |
| JigMark | 100.0% | 98.0% | 98.0% | 100.0% | 96.0% |
| PRC | 88.0% | 96.0% | 96.0% | 100.0% | 96.0% |
| StableSig | 100.0% | 100.0% | 100.0% | 100.0% | 100.0% |
| Trufo | 100.0% | 100.0% | 100.0% | 88.0% | 100.0% |
| GaussianShading + JigMark | 90.0% | 74.0% | 56.0% | 86.0% | 0.0% |
| StableSig + StegaStamp | 96.0% | 94.0% | 98.0% | 86.0% | 94.0% |

## J  Competition Resources

Official recordings on the competition, including descriptions and announcement of winners, are available at `https://neurips.cc/virtual/2024/competition/84795`. Further technical material

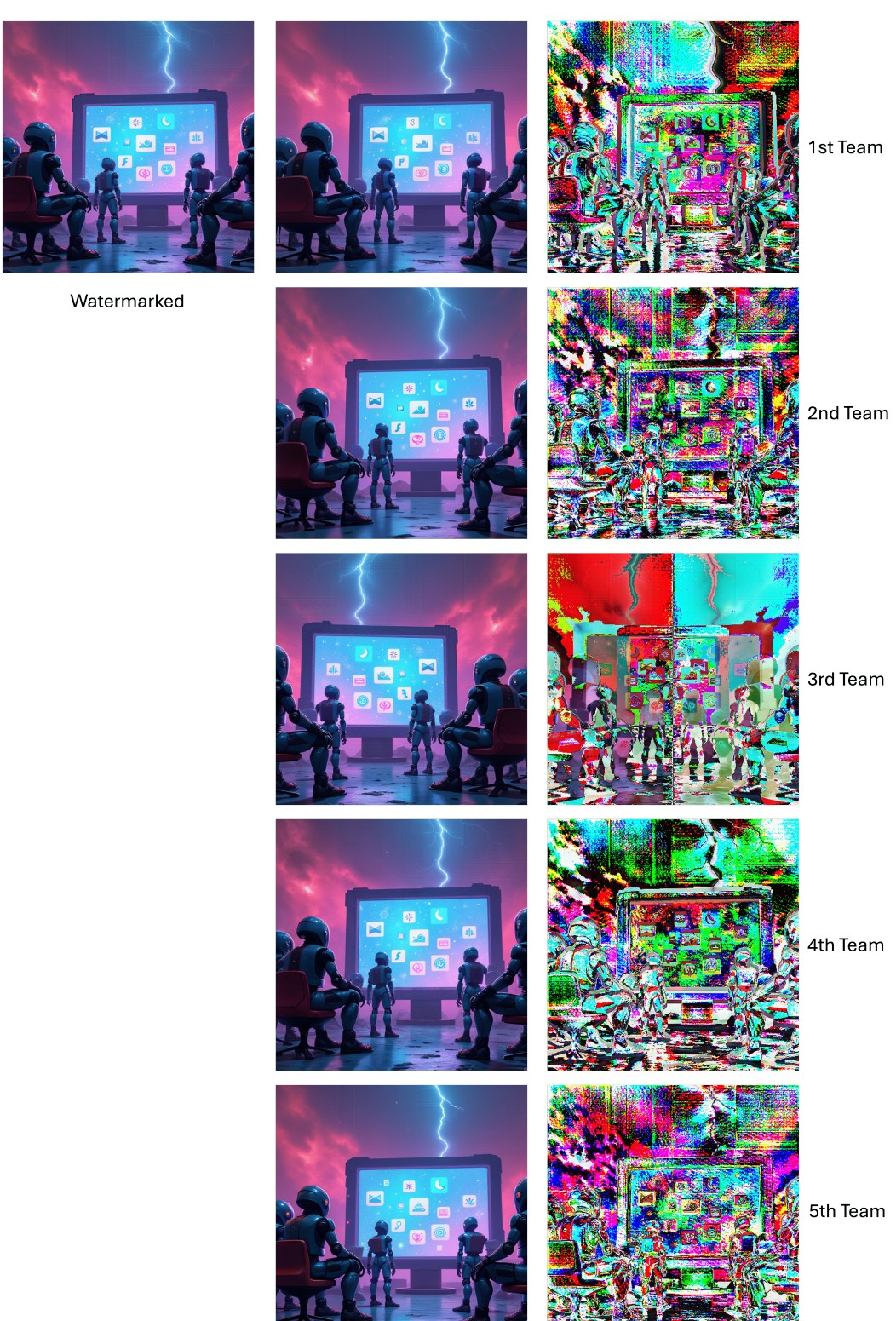

Figure 7: Examples of top 5 teams' attacks in the black-box track.

describing methodology and basis material is available at `https://erasinginvisible.github.io/`

