# OpenReview forum: "A Technical Report on “Erasing the Invisible”: The 2024 NeurIPS Competition on Stress Testing Image Watermarks"
_NeurIPS.cc/2025/Datasets_and_Benchmarks_Track — NeurIPS 2025 Datasets and Benchmarks Track poster_

### Official Review · Reviewer_kd4a · 2025-06-01

**Rating:** 4
**Confidence:** 3

**Summary:**

The authors organized “Erasing the Invisible,” a NeurIPS 2024 competition and newly established benchmark designed to systematically stress testing the resilience of watermarking techniques. The competition introduced two attack tracks—Black-box and Beige-box.

**Dataset Code Accessibility:**

Yes

**Ethical Considerations:**

No, there are no or only very minor ethics concerns

**Final Justification:**

The rebuttal addresses most of my concerns. I keep my rating.

**Limitations Weaknesses:**

1. More combinations of in-processing watermarking and post-processing watermarking should be considered. With enumeration of all possible combinations, the experimental results could be more convincing and justify the proposed conclusions.
2. More quantitative results should be provided, besides the box plot and scatter plot. This paper should list a table to summarize all evaluation metrics (e.g., watermark detection rate, image quality degradation) in terms of various types of watermarking.
3. More discussions should be provided on the trade-off between watermark detection performance and image quality degradation. Is the trade-off between them inevitable and why? Is there any way to circumvent such trade-off?

**Strengths Contributions:**

1. The paper is overall well-written and well organized. The highlighted two pitfalls are critical problems in watermarking and need to be well addressed.
2. The benchmark is carefully designed with different in-processing watermarks nd post-processing watermarks. Both beige-box and black-box are considered.
3. This paper points out the remaining problems of modern watermarks, and provide the insights from the winning beige-box attacks and winning black-box attacks. The summarized lessons and open questions are valuable.

---

> ### Author Rebuttal · Authors · 2025-07-31
>
> Thank you for your valuable suggestions and questions! Below, we address your questions and concerns in detail.
>
> >**W1**: More combinations of in-processing watermarking and post-processing watermarking should be considered. With enumeration of all possible combinations, the experimental results could be more convincing and justify the proposed conclusions.
>
> **Answer to W1**:
>
> Thanks for this suggestion -- indeed it is entirely possible that other combinations of watermarks could prove to be stronger than all combinations studied. We wanted to limit the total dataset size while still representing each combination with a sufficient batch size (in our case, 50). The combinations we proposed were watermarks with similar methodologies -- such as Stegastamp+Stable Signature (region-independent embedding) and Gaussian Shading+JigMark (region-targeted embedding) because we believed this would produce a synergistic boost in robustness. Our ultimate goal was to discover whether a combination could improve robustness, and indeed the GS+JigMark watermark was the most resilient, encouraging future work on combined methods.
>
> >**W2**: More quantitative results should be provided, besides the box plot and scatter plot. This paper should list a table to summarize all evaluation metrics (e.g., watermark detection rate, image quality degradation) in terms of various types of watermarking.
>
> **Answer to W2**:
>
> Thank you for raising this point. We fully agree that clearly presenting our evaluation metrics is important, and we appreciate the request for more quantitative detail. The original evaluation metrics for each submissions are all included in the subset "Beige_Track_Submissions" and "Black_Track_Submissions" of our released dataset on Hugging Face. Due to space constraints, these details were primarily documented in the supplementary material, but we summarize them here and map them to the corresponding column names in the dataset.
>
> Our competition rigorously assessed the trade-off between watermark removal efficacy and image quality preservation, quantified by a final score calculated as the Euclidean distance: $Score=\sqrt{Q^2+A^2}$
>
> Here's a breakdown of the key metrics:
>
> **Watermark Detection Performance (A)**: This metric reflects the watermark removal rate, defined as 1 - TPR@0.1%FPR. A True Positive Rate (TPR) was determined at a stringent False Positive Rate (FPR) of 0.1%. A detection threshold was established by analyzing 10,000 unwatermarked images, with an image considered "detected" if its decoding score fell below this pre-calculated 0.1 percentile threshold. An 'A' value of 1 indicates complete watermark removal, while 0 indicates no removal.
>
> | Column name | Evaluation metrics |
> | -------- | -------- |
> | gaussianshading     | TPR@0.1%FPR of all images watermarked by GaussianShading   |
> | jigmark     | TPR@0.1%FPR of all images watermarked by JigMark (only applicable for Black Track)  |
> | prc     | TPR@0.1%FPR of images watermarked by PRC (only applicable for Black Track) |
> | stablesig     | TPR@0.1%FPR of all images watermarked by StableSignature (only applicable for Black Track) |
> | stegastamp    | TPR@0.1%FPR of all images watermarked by StegaStamp   |
> | trufo    | TPR@0.1%FPR of all images watermarked by Trufo (only applicable for Black Track)|
>
> **Normalized Image Quality Degradation (Q)**: This score quantifies the visual impact of attacks by aggregating information from eight distinct image quality metrics (IQMs). These IQMs include PSNR, SSIM, NMI, FID, CLIP-FID, LPIPS, Delta Aesthetics Score, and Delta Artifacts Score, comparing the attacked image to its original watermarked version. The IQM scores were normalized based on empirically derived percentiles from a large corpus of images subjected to various attacks, and Q is a weighted average of these normalized scores. A higher 'Q' value signifies greater image degradation.
>
> | Column name | Evaluation metrics |
> | -------- | -------- |
> | aesthetics | Aesthetics score [1] of all images |
> | artifacts | Artifacts score [1] of all images |
> | legacy_fid |  Frechet Inception Distance (FID) [2] of all images |
> | clip_fid | CLIP feature space (CLIP-FID) [3] of all images |
> | lpips | Learned Perceptual Image Patch Similarity (LPIPS) [4] of all images |
> | nmi | Normalized Mutual Information (NMI) of all images |
> | psnr | Signal-to-Noise Ratio (PSNR) of all images|
> | ssim | Structural Similarity Index (SSIM) of all images|
>
> The objective for participants was to minimize the overall score, representing successful watermark removal (A≈0) with minimal impact on image quality (Q≈0). More detailed information, including specific formulas and normalization procedures, can be found in Section E.2 of our supplementary material.
>
> | Column name | Evaluation metrics |
> | -------- | -------- |
> | performance | The watermark removal socre of each submission $$A = 1− \text{TPR}@0.1\%\text{FPR}$$|
> | quality | The image quality score of rach submission $$\begin{aligned}Q =&  + 1.53 × 10^{−3} \text{FID} + 5.07 × 10^{−3} \text{CLIP FID}− 2.22 × 10^{−3} \text{PSNR} − 1.13 × 10^{−1} \text{SSIM} \\\\ & − 9.88 × 10^{−2}\text{NMI} + 3.41 × 10−1 \text{LPIPS} + 4.50 × 10^{−2}\Delta\text{Aesthetics}− 1.44 × 10^{−1}\Delta\text{Artifacts}\end{aligned}$$ |
> | score | The overall score of each submission $$Score=\sqrt{Q^2+A^2}$$ |
>
> We will add a summary of the metrics to the main paper.
>
> >**W3**: More discussions should be provided on the trade-off between watermark detection performance and image quality degradation. Is the trade-off between them inevitable and why? Is there any way to circumvent such trade-off?
>
> **Answer to W3**:
>
> Thank you for raising an excellent point about the trade-off between watermark detection performance and image quality degradation. We address this crucial aspect in our paper, particularly in Section 3.1, under "Watermark Detection Performance versus Image Quality Degradation Trade-offs", and provide some further discussions here.
>
> 1. Is the trade-off inevitable and why?
> Our findings, as visualized by the Pareto frontiers in Figure 2b, strongly suggest a trade-off between watermark removal and image quality. In the context of our competition, achieving both optimal removal and high image quality proved to be a significant challenge for participants. This is primarily because our Normalized Image Quality Degradation (Q) metric, as detailed in Section 3.3, is designed to be highly sensitive to even subtle distortions on the image, making the preservation of visual quality a central and demanding aspect of our competition. We observed that aggressive watermark removal often led to noticeable image degradation, likely because altering the image to eliminate the embedded signal frequently impacts its content and quality.
> 2. Is there any way to circumvent such trade-off?
> While no submission in our competition demonstrated an attack method that effectively circumvents this trade-off, we cannot preclude the possibility that future research may discover such a method. Should such an attack emerge that achieves both low watermark detection performance and low image quality degradation, it would likely indicate a limitation in the current image degradation metrics. In such a scenario, an improved version of these metrics would be necessary to accurately assess the perceived quality of the attacked images.
>
>
>
> [1] Xu, Jiazheng, et al. "Imagereward: Learning and evaluating human preferences for text-to-image generation." Advances in Neural Information Processing Systems 36 (2023): 15903-15935.
>
> [2] Heusel, Martin, et al. "Gans trained by a two time-scale update rule converge to a local nash equilibrium." Advances in neural information processing systems 30 (2017).
>
> [3] Kynkäänniemi, Tuomas, et al. "The Role of ImageNet Classes in Fréchet Inception Distance." ICLR. 2023.
>
> [4] Zhang, Richard, et al. "The unreasonable effectiveness of deep features as a perceptual metric." Proceedings of the IEEE conference on computer vision and pattern recognition. 2018.

---

> > ### Comment · Reviewer_kd4a · 2025-08-05
> >
> > The rebuttal addresses most of my concerns. I keep my rating.

---

### Official Review · Reviewer_m1vZ · 2025-07-02

**Ethics Flags:** Safety and security
**Rating:** 4
**Confidence:** 3

**Summary:**

This paper presents a detailed technical report on the Erasing the Invisible competition, which aimed to systematically evaluate the robustness of mainstream watermarking techniques against real-world attacks. The authors provide a thorough multi-dimensional analysis of the competition results, offering valuable insights into the current security of watermarking methods.

**Dataset Code Accessibility:**

Yes

**Dataset Code Comments:**

The authors provide accessible links where the dataset, benchmark data, and code can be downloaded. The materials are well-documented, available in a usable format, and sufficient for reproducibility.

**Ethical Comments:**

The paper provides detailed descriptions of the winning attack method, which could potentially be misused to remove watermarks in unauthorized or malicious contexts.

**Ethical Considerations:**

Yes, there are ethics concerns that require attention by the authors

**Final Justification:**

After reading the rebuttal, other reviewers’ comments, and re-evaluating the paper, I’ve decided to increase my score to 4. The authors clarified key implementation details, addressed concerns about evaluation, and overall demonstrated that the work is solid and valuable.

**Limitations Weaknesses:**

1. The work does not yet constitute a complete benchmark. Although the competition established an initial comparison framework based on attack success rates and image quality metrics, the lack of a unified input-output interface and standardized evaluation protocol limits its value as a consistent benchmark for future watermarking methods.

2 . The appendix includes a large number of high-resolution images, resulting in a bulky file size. Compressing images in future versions would improve readability and ease of access.

**Strengths Contributions:**

1. The report systematically summarizes the performance of various watermarking methods under different attack conditions, covering both Black-box and Beige-box scenarios, and highlights their vulnerabilities in practical settings.

2. The experiments on watermark superposition are particularly noteworthy, demonstrating that combining different watermarking techniques can significantly enhance robustness and offer a promising direction for future watermark design.

3. The authors analyze the attack strategies of winning teams and derive useful guidance for improving watermarking techniques.

---

> ### Author Rebuttal · Authors · 2025-07-31
>
> Thank you for your valuable suggestions and questions! Below, we address your questions and concerns in detail.
> >**W1**: The work does not yet constitute a complete benchmark. Although the competition established an initial comparison framework based on attack success rates and image quality metrics, the lack of a unified input-output interface and standardized evaluation protocol limits its value as a consistent benchmark for future watermarking methods.
>
> **Answer to W1**:
>
> Thank you for raising this concern. We respectfully clarify that our work does provide a complete and ready-to-use benchmark.
>
> We have publicly released three key assets:
> 1. A Standardized Evaluation Toolkit as a containerized program.
> 2. The Full Competition Dataset, including all watermarked images.
> 3. The Complete Set of 2,722 Participant Attack Submissions.
>
> This framework offers a unified interface that allows researchers to easily test new attacks against our watermarked dataset. Following the instructions in our GitHub repository, users can submit attacked images (input) and receive evaluation scores (output), computed by the standardized evaluation toolkit. Importantly, the toolkit is watermark-agnostic and can also be used to benchmark new watermarking methods. For example, researchers can apply existing attacks (including baseline attacks from WAVES [1] and winning solutions from this competition) to a new watermark, then evaluate the attack success using our toolkit to assess its robustness.
>
> If we have misunderstood the reviewer’s concern, we would be grateful for further clarification.
>
> [1] An, Bang, et al. "Waves: Benchmarking the robustness of image watermarks." arXiv preprint arXiv:2401.08573 (2024).
>
> >**W2**: The appendix includes a large number of high-resolution images, resulting in a bulky file size. Compressing images in future versions would improve readability and ease of access.
>
> **Answer to W2**:
>
> Thank you for this practical suggestion. We will compress the images in the appendix for faster loading.
>
> >**Ethical Comments**: The paper provides detailed descriptions of the winning attack method, which could potentially be misused to remove watermarks in unauthorized or malicious contexts.
>
> **Answer to Ethical Comments**:
>
> Thanks for bringing it up. This study, while reveals vulnerabilities in modern image watermarks, the goal is to increase awareness and accelerate the development of more robust watermarks. Exposing weaknesses ahead of time helps researchers design stronger watermarks. We include enough mechanism for verification, but we do not release a removal tool. Attacked outputs are limited to benchmark images under a research license, and methods are tailored to our specific benchmarked watermark families. We will add a short Ethical Use note to the paper and data, reiterating that our artifacts are for research and defense only.

---

> > ### Comment · Reviewer_m1vZ · 2025-08-08
> >
> > The author addressed my concerns in rebuttal, I have raised my score.

---

### Official Review · Reviewer_AEsU · 2025-07-02

**Rating:** 5
**Confidence:** 4

**Summary:**

This paper summarizes and analyzes the results in the competition "Erasing the Invisible: The 2024 NeurIPS Competition on Stress Testing Image Watermarks", which include two tracks: Beige-box Track and Black-box Track. The results of the competition are comprehensively analyzed and several important intuitions are provided.

**Dataset Code Accessibility:**

Yes

**Dataset Code Comments:**

The original watermarked image datasets and the complete set of participant submissions with
detailed evaluation results are all publicly released in the link provided in the page 4.

**Ethical Considerations:**

No, there are no or only very minor ethics concerns

**Final Justification:**

Thank you to the author for the rebuttal and considering my suggestions. I'll keep the score of 5.

**Limitations Weaknesses:**

The table on page 7 needs to be rotated for proper viewing. Optimizing the layout would improve readability.

This paper talks about general insights in the main text. However, I feel it better to provide a more detailed introduction and deeper understanding of the most effective attacks in the main text to help us understand the vulnerability of the current watermarking scheme better, instead of briefly mentioning it in section 3.3 and section 4.1.

**Strengths Contributions:**

The paper is well-written, organized and very easy to understand.

The robustness of image watermarking is a very important research question. Nowadays, with the rapid spread of very strong generative models and increasing requirements from both policy and regulation, image watermarking has become a widely used tool for distinguishing AI-generated images from real ones, as well as for tracing the source of AI-generated content. This paper is therefore very timely and I do not know many similar works before.

There are abundant submissions (2,722 submissions from 65+77 teams) in this competition, making the results very reliable.

The insights drawn from the results are very interesting and intuitive: the Pareto frontier around a watermark detection performance level approximately 0.5; the trade-off between watermark detection performance and image quality degradtion; advantages of watermark superposition are very interesting and meaningful. Also, the competition results highlight the fact that current imagine watermark is far from perfect in the aspect of robustness.

---

> ### Author Rebuttal · Authors · 2025-07-31
>
> Thank you for your valuable suggestions and questions! Below, we address your questions and concerns in detail.
> >**W1**: The table on page 7 needs to be rotated for proper viewing. Optimizing the layout would improve readability.
>
> **Answer to W1**:
>
> Thanks for the suggestion. We will refine the layout.
>
> >**W2**: This paper talks about general insights in the main text. However, I feel it better to provide a more detailed introduction and deeper understanding of the most effective attacks in the main text to help us understand the vulnerability of the current watermarking scheme better, instead of briefly mentioning it in section 3.3 and section 4.1.
>
> **Answer to W2**:
>
> This is a great suggestion. In Appendix I, we introduced all the winning solutions, which are the most effective attacks found through this competition, in detail, and cited their corresponding papers/blogs/code. We will add more information on the effective attacks and the insights we discovered, and discuss the vulnerability of current watermarks in Section 3.

---

> > ### Comment · Reviewer_AEsU · 2025-08-05
> >
> > Thank you for the rebuttal and considering my suggestions. I'll keep the score of 5.

---

### Official Review · Reviewer_DTMx · 2025-07-07

**Rating:** 5
**Confidence:** 4

**Summary:**

The paper is a report on the 2024 NeurIPS competition on removal attacks on image watermarks. The authors release all the target images, teams' submissions, and the infrastructure code used to run the competition, and discuss the resulting insights.

**Additional Feedback:**

Some typos:
- L8: testing->test
- L215: robustness->robust

Q: L79 states all watermarks were tuned to 0.1% FPR and had 99.9% TPR. It is not clear to me how this is possible? From my understanding most methods have a threshold that can be set to achieve the desired (empirical) FPR. But the TPR at this point would then be different depending on the method?

Q: I like the idea of a 2D score incorporating both quality and strength, but my understanding of attack success in removal attacks was mostly along the lines of "minimize TPR while keeping quality above some threshold"; did the authors consider this as the alternative and can they argue for their choice? This discussion would also be beneficial in the paper.

**Dataset Code Accessibility:**

Yes

**Dataset Code Comments:**

The dataset and all code used to run the competition are open-sourced and seem well-documented. I did not attempt to set up and run the evaluation logic myself.

**Ethical Considerations:**

No, there are no or only very minor ethics concerns

**Final Justification:**

The authors have answered my questions in the rebuttal and promised to extend the dataset to mitigate the discussed issues. My score remains "Accept" as I believe the paper is a good addition to the conference.

**Limitations Weaknesses:**

- The quality metric while elaborate and robust, is hard to interpret. This is particularly problematic when trying to interpret Figure 2b: what does it mean to have 0.1 quality score? Is this, as in WAVES, based on empirical CDF of a set of attacks? The authors should also more clearly indicate which part of quality evaluation is novel compared to WAVES which seems to share most of the design choices. An additionally interesting but perhaps out of scope step would be to see if the metric can be hacked, i.e., can we find images that clearly have human-visible artifacts but score highly on the metric. I do not consider this as a strong weakness, but this type of analysis would make the work much stronger as a benchmarking tool.
- The value of releasing the submissions without images is not clear to me on top of the analysis done by the authors already. Do the authors have ideas how this data can be useful in new ways?
- I did not notice the code used to generate the dataset. This is in my opinion important to release as implementation differences in watermarks can cause different outcomes and this can be hard to debug for someone trying to e.g. extend this benchmark.
- The dataset does not seem to include labels mapping each image to the watermark(s) used. This can be inferred from submission data but should be present on the top level as part of the release.

**Strengths Contributions:**

- The work is a significant contribution as it improves on the state of genai watermark benchmarking. In particular, as the authors also discuss, this is the first large-scale study of image watermark removal attacks against real-world attacks, as opposed to prior benchmarks like WAVES which only focus on a set of hardcoded transformations.
- Given notable participation in the competition the insights obtained this way seem significant and authors do a good job to discuss them. Table 2 is quite informative and high-level patterns identified can inform future research: success of unexpectedly simple geometric attacks, postprocessing needed to recover quality after regeneration attacks, value of hiding the watermark algorithm as tailored attacks are more successful, difficulty of scrubbing complementary combinations of watermarks, etc.
- There is additional value in releasing all the infrastructure needed to run the competition, as this makes it easier to run similar competitions or live benchmarks in the future.

---

> ### Author Rebuttal · Authors · 2025-07-31
>
> Thank you for your valuable suggestions and questions! Below, we address your questions and concerns in detail.
> > **W1**: The quality metric while elaborate and robust, is hard to interpret. This is particularly problematic when trying to interpret Figure 2b: what does it mean to have 0.1 quality score? Is this, as in WAVES, based on empirical CDF of a set of attacks? The authors should also more clearly indicate which part of quality evaluation is novel compared to WAVES which seems to share most of the design choices. An additionally interesting but perhaps out of scope step would be to see if the metric can be hacked, i.e., can we find images that clearly have human-visible artifacts but score highly on the metric. I do not consider this as a strong weakness, but this type of analysis would make the work much stronger as a benchmarking tool.
>
> **Answer to W1**:
>
> Thank you for the question.Our quality metric Q is indeed adopted directly from the WAVES benchmark. We chose this metric because it is already a robust and well-established standard that aggregates eight Image Quality Metrics (IQMs).
>
> To clarify its interpretation: a score of Q=0.1 corresponds to the 10th percentile of image degradation observed across a large set of attacks from the WAVES benchmark. This provides a standardized and interpretable scale.
>
> Regarding the metric being "hacked," we performed human verification of the top submissions. The automated Q score aligned closely with human evaluation of image quality, and we found no instances of high scores for images with obvious artifacts. This gives us confidence in its robustness.
>
> Our contribution is not a new metric, but rather the first large-scale, real-world stress test of this state-of-the-art evaluation framework. We will clarify the metric's origin and interpretation in the final version.
>
> > **W2**: The value of releasing the submissions without images is not clear to me on top of the analysis done by the authors already. Do the authors have ideas how this data can be useful in new ways?
>
> **Answer to W2**:
>
> Our dataset also provides attacked images from every submission. The "How to Use" section in our Hugging Face dataset card introduces how to access it. Thanks for your question. We will make it more clear in the dataset card.
>
> Our dataset contains all the original watermarked images of two tracks, and all the attacked images and evaluation details of ~2700 submissions. It is invaluable for researchers and practitioners in digital watermarking, adversarial machine learning, image forensics, and content authenticity. It provides a large-scale benchmark and rich baselines for:
>
> - Analyzing watermark vulnerabilities and successful attack strategies.
> - Developing and testing more robust watermarking defenses.
> - Benchmarking new watermark removal algorithms.
> - Research into image quality assessment under adversarial manipulations.
>
> > **W3**: I did not notice the code used to generate the dataset. This is in my opinion important to release as implementation differences in watermarks can cause different outcomes and this can be hard to debug for someone trying to e.g. extend this benchmark.
>
> **Answer to W3**:
>
> We thank the reviewer for raising this important point about releasing the code used for dataset generation. We fully agree that this is essential for reproducibility, debugging, and extending the benchmark.
>
> To clarify, the code used for dataset generation consists primarily of publicly available implementations of the image generation models (Stable Diffusion 2.1, Flux.1 [dev]) and the watermarking methods (three in-processing and three post-processing) that we employed. All of these implementations are open-sourced, with the exception of Trufo, which we accessed via an API provided by its developer.
>
> We understand the policy of the rebuttal process prevents us from updating our GitHub repository at this moment. We will add all the specific code and relevant configurations used for dataset generation to our repository after the review process concludes.
>
> >**W4**: The dataset does not seem to include labels mapping each image to the watermark(s) used. This can be inferred from submission data but should be present on the top level as part of the release.
>
> **Answer to W4**:
>
> We thank the reviewer for the excellent suggestion to include labels mapping each image to its watermark(s). We agree this would make the dataset significantly more convenient and user-friendly.
>
> We confirm that we have a complete record of the watermarks associated with each image, so there are no technical barriers to adding these labels to the released dataset. However, according to the policy, we are not allowed to update this information during the reviewing process. We will update our dataset with these comprehensive labels after the reviewing process.
>
> >**Additional Feedback 1**: L79 states all watermarks were tuned to 0.1% FPR and had 99.9% TPR. It is not clear to me how this is possible? From my understanding most methods have a threshold that can be set to achieve the desired (empirical) FPR. But the TPR at this point would then be different depending on the method?
>
> **Answer to AF1**:
>
> Thanks for raising this point of confusion. We only calibrated the FPR and found that all watermarks achieve a TPR of at least 99.9% on un-attacked watermarked images. Here are the details:
>
> 1. For each watermark decoder, we scanned its raw detection scores (which are continuous) on 10,000 unwatermarked images, selected the score that yields a 0.1% FPR as the threshold, and then fixed that threshold for the rest of the evaluation.
> 2. With the detection threshold fixed, we found that all watermarking methods on un-attacked watermarked images achieved a TPR of at least 99.9% without any tuning.
>
> We will update that sentence in L79 and clarify the process in the paper.
>
> >**Additional Feedback 2**: I like the idea of a 2D score incorporating both quality and strength, but my understanding of attack success in removal attacks was mostly along the lines of "minimize TPR while keeping quality above some threshold"; did the authors consider this as the alternative and can they argue for their choice? This discussion would also be beneficial in the paper.
>
> **Answer to AF2**:
>
> "Minimize TPR while keeping quality above some threshold" is a good idea. But we chose a continuous, two‑dimensional evaluation with a single scalarized score for two main reasons:
> 1. We want to avoide an arbitrary quality threshold. A fixed quality threshold depends on image and watermark which is hard to find a proper one. Small changes can also flip rankings and incentivize “gaming” just above the cutoff. Our score instead evaluates every submission at its actual (Q,A) and ranks by distance to the origin, rewarding balanced progress in both quality and removal.
> 2. We want a faithful comparison across the full Pareto frontier. Empirically, solutions that push only one axis at the extreme expense of the other are sub‑optimal. Our metric encourages attacks that simultaneously achieve low A and low Q.
>
> Therefore, we believe 2D score is a better choice. Thanks for the suggestion. We will add this discussion to the paper.

---

### Decision · Program_Chairs · 2025-09-18

**Decision:**

Accept (poster)

**Comment:**

This paper presents the NeurIPS 2024 competition “Erasing the Invisible”, which tested how well current watermarking methods hold up against real-world removal attacks in both Black-box and Beige-box settings, with strong participation. The work stands out as the first large-scale benchmark of its kind, going beyond hand-crafted transformations and releasing not just the data but also the submissions and infrastructure, making it a valuable resource for the community. Reviewers praised the clarity, scale, and timeliness of the paper, as well as the useful insights it provides—like the success of simple geometric attacks, the benefits of combining watermarks, and the trade-off between robustness and image quality. Some areas could be improved, such as clearer explanation of the quality metric, deeper discussion of top attacks, and providing dataset annotations, but these are relatively minor compared to the strengths. Overall, this is an impactful, well-executed benchmark that will shape future work on watermarking, and I recommend acceptance.

===== FINAL UPDATE FROM DB Track PCs ====

The final decision for this paper has been taken by the program chairs after consultation with the SACs. All Senior Area Chairs have ranked papers according to the feedback from the AC during the review process. We decided to leave the original meta-review to reflect the opinion of the AC in light of the initial discussions with reviewers and SAC.